# Inhibitory Effects of IL-6-Mediated Matrix Metalloproteinase-3 and -13 by *Achyranthes japonica* Nakai Root in Osteoarthritis and Rheumatoid Arthritis Mice Models

**DOI:** 10.3390/ph14080776

**Published:** 2021-08-07

**Authors:** Xiangyu Zhao, Dahye Kim, Godagama Gamaarachchige Dinesh Suminda, Yunhui Min, Jiwon Yang, Mangeun Kim, Yaping Zhao, Mrinmoy Ghosh, Young-Ok Son

**Affiliations:** 1Interdisciplinary Graduate Program in Advanced Convergence Technology and Science, Jeju National University, Jeju-si 63243, Korea; zhaoxiangyu@jejunu.ac.kr (X.Z.); godagama@jejunu.ac.kr (G.G.D.S.); reinise4011@jejunu.ac.kr (Y.M.); 2Department of Animal Biotechnology, Faculty of Biotechnology, Jeju National University, Jeju-si 63243, Korea; dahyekim@jejunu.ac.kr (D.K.); yangjw1127@jejunu.ac.kr (J.Y.); aksxhs123@jejunu.ac.kr (M.K.); 3Frontiers Science Center for Transformative Molecules, School of Chemistry and Chemical Engineering, Shanghai Jiao Tong University, Shanghai 200240, China; ypzhao@sjtu.edu.cn; 4KIIT-Technology Business Incubator (KIIT-TBI), Bhubaneswar 751 024, India; 5Bio-Health Materials Core-Facility Center, Jeju National University, Jeju-si 63243, Korea; 6Practical Translational Research Center, Jeju National University, Jeju-si 63243, Korea

**Keywords:** *Achyranthes japonica* Nakai root, osteoarthritis, rheumatoid arthritis, destabilization of the medial meniscus, collagenase-induced arthritis

## Abstract

*Achyranthes japonica* Nakai root (AJNR) is used to treat osteoarthritis (OA) and rheumatoid arthritis (RA) owing to its anti-inflammatory and antioxidant effects. This study investigated the inhibitory effects of AJNR on arthritis. AJNR was extracted using supercritical carbon dioxide (CO_2_), and its main compounds, pimaric and kaurenoic acid, were identified. ANJR’s inhibitory effects against arthritis were evaluated using primary cultures of articular chondrocytes and two in vivo arthritis models: destabilization of the medial meniscus (DMM) as an OA model, and collagenase-induced arthritis (CIA) as an RA model. AJNR did not affect pro-inflammatory cytokine (IL-1β, TNF-α, IL-6)-mediated cytotoxicity, but attenuated pro-inflammatory cytokine-mediated increases in catabolic factors, and recovered pro-inflammatory cytokine-mediated decreases in related anabolic factors related to in vitro. The effect of AJNR is particularly specific to IL-6-mediated catabolic or anabolic alteration. In a DMM model, AJNR decreased cartilage erosion, subchondral plate thickness, osteophyte size, and osteophyte maturity. In a CIA model, AJNR effectively inhibited cartilage degeneration and synovium inflammation in either the ankle or knee and reduced pannus formation in both the knee and ankle. Immunohistochemistry analysis revealed that AJNR mainly acted via the inhibitory effects of IL-6-mediated matrix metalloproteinase-3 and -13 in both arthritis models. Therefore, AJNR is a potential therapeutic agent for relieving arthritis symptoms.

## 1. Introduction

Osteoarthritis (OA) is a severe chronic degenerative disease of the joints that is common in middle aged and older people [1]. The main clinical manifestations of OA are degeneration of the articular cartilage and changes in the subchondral bone structure [2]. When joint cartilage is completely lost following disruption of cartilage homeostasis through the induction of catabolic factors as well as downregulation of anabolic factors, the bones and soft tissue structures around the joint are altered, resulting in joint pain, swelling, deformity, and disability [3,4]. Although several risk factors associated with OA have been proposed, including mechanical impairment, genetic factors, aging, obesity, sex, and metabolic diseases, the pathogenesis of OA is not fully understood. The current treatment of this disease is mainly aimed at relieving OA symptoms, such as pain relief, anti-inflammatory effects, and prevention of damage to the joint structure [5,6]. Chondrocytes produce various cytokines and chemokines, and respond to them in a paracrine or autocrine manner in joint tissues or synovial fluid [7,8]. The relationship between elevated levels of catabolic enzymes and inflammatory mediators (e.g., prostaglandins and nitric oxide) in OA synovial fluid and joint tissue has been well studied [9,10,11,12,13]. IL-1β, IL-6, and TNF-α are well-known pro-inflammatory cytokines involved in arthritis pathogenesis [2]. It has been reported that some particular transcription factors upregulate pro-inflammatory cytokines, e.g., endothelial PAS domain protein 1 (Epas1) [2], estrogen-related receptor gamma (Esrrg) [14], nicotinamide phosphoribosyl-transferase (Nampt) [15], metal regulatory transcription factor 1 (Mtf1) [16], runt-related transcription factor 2 (RUNX2) [17], basic leucine zipper transcription factor, ATF-like (BATF) [18], and RAR-related orphan receptor α (RORα) [19]. Previous studies have indicated that HIF-2α is a central player in OA pathogenesis, and that its target transcription factors, or non-transcription factors, are positive regulators of catabolic factors in OA pathogenesis [15,20,21]. In addition, toll-like receptor (TLR) signaling [22] as well as arginine [23], selenium [24], and cholesterol metabolisms [19] are associated with pro-inflammatory cytokines in OA pathogenesis. However, the mechanisms that trigger the production of inflammatory mediators remain unclear [8].

OA clinical treatment is mainly performed using two methods: joint cavity injection and oral medication [25]. Glucocorticoid or sodium hyaluronate is commonly administered via injection, whereas amino grape sulfate, non-steroidal anti-inflammatory drugs (NSAIDs), and opioids are administered orally. High doses of these drugs can cause many adverse reactions, such as gastrointestinal irritation and other apparent side effects, which can cause ulcers and perforations in severe cases [26]. Liver and kidney damage caused by arthritis medication can also induce adverse effects such as skin rash, urticaria, headache, dizziness, and drowsiness, with some patients even showing symptoms such as hypertension and edema after arthritis medication [27]. Thus, it is important to find alternative effective medicines with low side effects.

In recent years, supercritical fluid extraction has received a great deal of attention in the pharmaceutical industry [28,29]. Supercritical fluid technology has been applied to the production of food, flavoring, nutrients, and bioactive components from plant sources [30,31]. CO_2_ is the most ideal solvent for the supercritical extraction of natural products, as supercritical CO_2_ has low critical parameters, as well as hydrophobic and nonpolar characteristics [32]. Moreover, the ambient critical temperature makes it suitable for the extraction of a thermolabile component without degradation [32]. The CO_2_ supercritical fluid extraction method has been recognized as an environmentally friendly technology because it is completely free from the use of potentially toxic chemical solvents [28,29].

As a traditional natural component in many Asian countries, AJNR contains many types of compounds, including polysaccharides, saponins, sterones, flavonoids, polypeptides, organic acids, and various trace elements. These components have been used in OA treatment and have been shown to promote the proliferation of chondrocytes, reduce joint swelling, and inhibit synovial hyperplasia [33,34,35,36]. However, the underlying mechanisms of the inhibitory effects of AJNR on arthritis have not been elucidated.

This study explored the remission effects of AJNR against OA and RA. First, the AJNR compounds were extracted using supercritical CO_2_. This extraction method is unique and does not require the use of toxic solvents. Next, we examined the ameliorative effect of AJNR on arthritis using the primary culture of articular chondrocytes, and two in vivo model systems. The findings of this study will confirm whether the natural compound AJNR is a potential therapeutic compound that can inhibit arthritis pathogenesis without any side effects.

## 2. Results

### 2.1. Extraction and Identification of AJNR Components

The AJNR components were extracted using a supercritical CO_2_ extraction method (Figure 1a). The extraction pressure and temperature were set to 40 MPa and 60 °C, respectively. Because we used ethanol as a co-solvent during supercritical CO_2_ extraction and an ethanol soluble phase of AJNR extract, in our experiment, ethanol was used as a solvent for gas chromatography-mass spectrometry (GC-MS) analysis. The results showed that the main compounds of AJNR were pimaric acid (74%) and kaurenoic acid (26%) (Figure 1b).

### 2.2. Effects of AJNR on Cell Viability

The effects of AJNR on chondrocyte viability were evaluated by MTT assay. Treatment with AJNR (0–100 μg/mL) for 48 h did not result in a significant cytotoxic effect (Figure 2a). Moreover, following treatment with AJNR (10, 20, 50 μg/mL) in the absence or presence of the pro-inflammatory cytokines IL-1β (1 ng/mL), IL-6 (100 ng/mL), and TNF-α (10 ng/mL), the cell viability of primary chondrocytes was not recovered (Figure 2b–d). These results suggest that AJNR does not affect cell viability, either as a single treatment or in the presence of pro-inflammatory cytokines.

### 2.3. Effects of AJNR on Anabolic or Catabolic Factors in Primary Cultured Articular Chondrocytes

We analyzed the effects of AJNR on anabolic or catabolic factors in primary cultured articular chondrocytes. RT-PCR analyses revealed that AJNR (0–50 μg/mL) did not affect the expression of key OA inducer genes, such as Epas1 [2], solute carrier family 39 member 8 (Slc39a8) [20], Esrrg [14], Nampt [15], and Mtf1 [16] (Appendix A). In addition, AJNR had almost no effect on the expression of MMP-2, -3, -8, -9, -10, -12, -13, -14, and -15 (Appendix A). Moreover, the expression of the anabolic factors collagen type II alpha 1 chain (Col2a1), Aggrecan, and SRY-box transcription factor 9 (Sox9), as well as the catabolic factors ADAM metallopeptidase with thrombospondin type 1 motif (Adamts)-4 and -5, was not affected by AJNR (Appendix A).

### 2.4. Effects of AJNR on Chondrocytes Exposed to Pro-Inflammatory Cytokines

Treatment with pro-inflammatory cytokines, namely IL-1β (0, 0.1, 0.5, 1 ng/mL), TNF-α (0, 1, 5, 10 ng/mL), and IL-6 (0, 10, 50, 100 ng/mL), increased catabolic factors (Mmp3, Mmp10, Mmp13, and Adamts5) and decreased anabolic factors (Col2a1, Sox9, and Aggrecan) remarkably in mouse primary cultured chondrocytes in a dose-dependent manner (Figure 3a–c). However, AJNR attenuated the increases in catabolic factors induced by IL-1β, TNF-α, and IL-6. AJNR also recovered IL-1β-, TNF-α-, and IL-6-induced decreases in anabolic factors (Figure 3d–f). Notably, AJNR showed a specific effect on IL-6-mediated anabolic and catabolic alterations (Figure 3f).

### 2.5. AJNR Attenuated DMM-Induced OA Pathogenesis

We further investigated the effect of supercritical CO_2_ extract of AJNR on OA pathogenesis in a DMM model. IP injection of AJNR was administered twice a week for 8 weeks after DMM surgery (Figure 4a). The structural integrity of the articular cartilage was evaluated via staining with Safranin-O/Fast Green (Figure 4b). The results showed that AJNR attenuated DMM-induced cartilage degeneration and erosion. The other OA parameters, such as OARSI (*p* < 0.0001), sclerosis (*p* = 0.005), and osteophytes (*p* < 0.0001), were completely blocked by AJNR in the DMM model (Figure 4c–f). In addition, DMM-induced synovial inflammation (as indicated by increased articular cavity inflammatory cells, thickened synovial membrane, and synovial tissue edema) was attenuated by AJNR (*p* < 0.0001) (Figure 4g,h). In both sham groups, the articular cartilage surface was smooth, and the synovial tissue was not hyperplastic.

### 2.6. AJNR Attenuated CIA-Induced RA

We determined whether AJNR has inhibitory effects on RA in a CIA model. IP injection of AJNR (2 mg/kg) was performed during CIA induction (Figure 5a). Histological assessments were conducted on mouse ankle, knee, and toe joints. There were no pathological symptoms of arthritis in the non-immunized (NI) group. However, severe synovitis with synovial hyperplasia, erosion of the bone and cartilage, and infiltration of inflammatory cells were observed in the CIA group (Figure 5b,e). CIA-induced cartilage degeneration and synovitis were dramatically reduced in the AJNR-treated group (Figure 5b,e). The histological scores of synovitis and OARSI were also significantly attenuated in the ankle, knee, and toe of mice treated with AJNR (Figure 5c,d,f–i). These results suggest that AJNR has beneficial effects against inflammation and cartilage degeneration in RA.

### 2.7. AJNR Attenuated CIA-Induced Pannus Formation in Ankle and Knee

Pannus formation is an essential indicator of severe arthritis. Therefore, we evaluated CIA-induced pannus formation in mouse ankles and knees. The results showed that pannus formation in the ankle and knee was not observed in the NI group; however, CIA-induced pannus formation was significantly reduced by AJNR in both the ankles (*p* = 0.0007) and knee (*p* = 0.0293) (Figure 6).

### 2.8. AJNR Inhibited IL-6 Mediated Mmp3 and Mmp13 Expression in DMM-Induced OA Model or CIA-Induced RA Model

Our in vitro mechanistic studies revealed that the inhibitory effects of AJNR were specific against IL-6. To further verify the inhibitory effects of AJNR on IL-6-mediated OA pathogenesis, we performed immunohistochemical staining of samples from the DMM model. As shown in Figure 7a,b, IL-6, Mmp3, and Mmp13 expression was elevated in both the knee cartilage and knee synovium of the DMM model, compared with that in the sham group; however, the elevated expression of these proteins was remarkably attenuated in the AJNR-treated group (Figure 7a,b). To further verify the mechanism of IL-6-mediated Mmp3 and Mmp13 expression, we examined the expression of IL-6, Mmp3, and Mmp13 in the CIA model. The expression of IL-6, Mmp3, and Mmp13 was dramatically enhanced in the ankle cartilage and synovium of CIA-induced RA mice (Figure 7c,d). Similar to that in the DMM model, the elevated expression of IL-6, Mmp3, and Mmp13 was reduced in the AJNR-treated CIA model (Figure 7c,d). These results indicate that the inhibitory effects of AJNR are specific for IL-6-mediated Mmp3 and Mmp13 expression.

## 3. Discussion

In this study, we investigated the effect of supercritical CO_2_ extract of AJNR against arthritis pathogenesis in vitro and in vivo. We used AJNR at a concentration of 50 μg/mL in the in vitro study on the basis of the results of the articular chondrocyte viability test (Figure 2). AJNR particularly inhibits catabolic alteration mediated by IL-6, among pro-inflammatory cytokines (i.e., IL-1β, TNF-α, and IL-6) (Figure 3). Based on our preliminary test of AJNR in animals, IP injection of AJNR was administered at a concentration of 2 mg/mL. IP injection of AJNR effectively protected and slowed down the pathogenesis of post-traumatic OA in mouse DMM and CIA models of RA. Compared with those in the arthritis group, cartilage erosion and proteoglycan loss, synovitis, and subchondral plate thickness were reduced in the AJNR treatment group. These phenomena are explicitly related to IL-6-mediated Mmp3 and Mmp13 expression. Taken together, our findings showed that the potential of AJNR for joint protection and arthritis treatment was significant.

In accordance with the OA concept put forward by Garrod in 1890, the pathological characteristics of OA in inflammatory lesions of the joints were evaluated [37]. The main pathological changes in OA are articular cartilage loss and collagen fiber degeneration [38], which can lead to bone hyperplasia and osteophyte formation when the original mechanical balance of the knee joint is disrupted [39]. Therefore, these pathological changes can reflect the clinical characteristics of this degenerative joint disease [40,41]. Mmps participate in the degradation of many matrix components, and the activities of Mmps are regulated by hormones and cytokines in vivo [20,42]. Mmps play an essential role in synthesizing and decomposing the matrix and in the intervention of many physiological and pathological processes, such as arthritis and tissue remodeling [42]. Our results showed that AJNR effectively inhibited IL-6-induced Mmp3, Mmp10, and Mmp13 expression, but this effect was not specific to IL-1β and TNF-α (Figure 3). ADAMTS4 and ADAMTS5 are the main proteases responsible for the degradation of proteoglycans in the articular cartilage of OA [43,44,45,46]. AJNR was only effective against IL-1β- or TNF-α-induced ADAMTS5 expression (Figure 3). In addition, AJNR recovered the reduction of SOX9 and Aggrecan mediated by all three pro-inflammatory cytokines (Figure 3).

There has been no evidence of the complete curation of OA. The main clinical treatments for this disease are oral drugs, such as NSAIDs, which can cause many adverse side effects [47,48,49]. Nevertheless, it is important to protect the joints of patients with the disease. The inhibitory effect of AJNR against OA symptoms indicates its potential as a therapeutic agent against OA; however, further validation in other animal models and clinical trials is required [50].

The saponins and sterones in AJN, which have antitumor and anti-inflammatory effects, are believed to be the main medicinal substances of AJN [51,52]. The flavonoids, alkaloids, allantoin, succinic acid, and β-sitosterol of AJN have also been evaluated for pharmacological activities [52]. So far, researchers have focused only on the extract ingredients of AJN, which possess inhibitory effects against arthritis [53,54]. Moreover, the animal model they used was not a post-traumatic model such as monosodium iodoacetate-induced osteoarthritis animal model [55] or rabbit CIA model [56]. Furthermore, there is a lack of available information on the active compounds of AJN. In this study, we identified two novel main components from AJNR, namely, pimaric acid and kaurenoic acid, extracted using CO_2_ supercritical fluid (Figure 1). Because there has been no report of these compounds from AJNR extraction via normal organic solvent extraction methods nor functional research, our current study guaranteed functional studies of these novel compounds in arthritis pathogenesis in the future. As a traditional plant medicine, AJNR has been used in the clinical treatment of OA for many years [52,57,58]. Our data indicated the inhibitory effects of AJNR on all OA parameters, e.g., Safranin-O-stained tissue histology, OARSI grade, subchondral bone plate thickness, osteophyte size, osteophyte maturity, and synovitis (Figure 4). Moreover, AJNR displayed an inhibitory effect against CIA-induced arthritis in the knee, ankle, and toe (Figure 5 and Figure 6). Finally, our data suggested that AJNR exerted a specific effect on IL-6-mediated alterations in Mmp3 and Mmp13 expression in OA and RA mouse models, as confirmed by immunohistochemistry analysis.

## 4. Materials and Methods

### 4.1. Chemicals and Laboratory Ware

Unless otherwise stated, chemicals and laboratory wares were purchased from Sigma Chemical Co. (St. Louis, MO, USA) and Falcon Labware (Becton-Dickinson, Franklin Lakes, NJ, USA). Dulbecco’s modified Eagle medium (DMEM), fetal bovine serum, and recombinant human IL-6 (PHC0064) were purchased from Gibco (Grand Island, NY, USA). Recombinant mouse TNF-α (Cat. No. Z02918-20) and human IL-1β (Cat. No. Z02922-10) were purchased from GeneScript (Piscataway, NJ, USA).

### 4.2. Preparation of AJNR Extracts by Supercritical CO_2_

The extracts for the experiment were prepared using supercritical CO_2_ extraction, which is widely applied to extract bioactive substances from natural materials [31,59]. The extraction was performed using a laboratory-scale supercritical CO_2_ extraction device (RZSCF130-65-01L SUPERCRITICAL CO2 EXTRACTION EQUIPMENT) manufactured by Nantong Wisdom Supercritical Science & Technology Development Co., Ltd., Haian, China, which essentially consisted of a CO_2_ source, a condenser, a CO_2_ storage tank, a CO_2_ pump, an extractor, and three separators, as shown in the schematic drawing in Figure 1a. During extraction, 500 g of dried and powdered AJNR (approximately 40 meshes) mixed with 250 mL of absolute ethanol were loaded into the extractor, in which the air was replaced with CO_2_ three times. The CO_2_ was pumped into the extractor, and the extracts dissolved in CO_2_ were transported out from the extractor and separated from CO_2_ at the separators. The extraction pressure and temperature were set to 40 MPa and 60 °C, respectively. The pressure and temperature of the separators were set as 8 MPa, 50 °C (separators 6-I), and 5 MPa, 40 °C (separators 6-II and 6-III). The CO_2_ flow rate was controlled at 20 kg/h, and the extraction time was 2 h. After being separated from the extract, the CO_2_ would return to the condenser and repeat the cycle. After the extraction was complete, CO_2_ was released to the atmosphere via an empty valve, the extracts were collected from the separators and further evaporated using a rotavapor to remove the ethanol. The final extract (approximately 10 g) was stored in a sealed airtight tube for subsequent tests.

### 4.3. Gas Chromatography-Mass Sepectrometry Analysis

AJNR compounds were analyzed using a Shimadzu GCMS QP2010 SE gas chromatography-mass spectrometer (Shimadzu, Kyoto, Japan). An HP-5 (25 m × 0.32 mm × 0.17 µm) column was used. The injector temperature was 250 °C, and the oven temperature was 80 °C–20 °C/min-200 °C (5 min)–250 (3 min). The column flow was 0.5 mL/min, and the injection volume was 1.0 µL.

### 4.4. Primary Culture of Mouse Knee Joint Chondrocytes and Treatment with AJNR

All animal experiments were approved by the Jeju National University Animal Care and Use Committee (2020-0001), approved on 6 February 2020. Chondrocytes were isolated from the femoral condyles and tibial plateaus of 4-day-old mice (*n* = 8) by digesting cartilage tissue with DMEM supplemented with 0.2% collagenase (Sigma). The passage “0” (P0) primary chondrocytes (3 × 10^5^/30 mm culture dish) were maintained as a monolayer in DMEM (Gibco, Waltham, MA, USA) supplemented with 10% fetal bovine serum and antibiotics (100 units/mL penicillin G and 100 μg/mL streptomycin; Gibco, Waltham, MA, USA) for 24 h in a 5% CO_2_ incubator at 37 °C. The chondrocytes were then exposed to various concentrations of AJNR (10, 20, and 50 μg/mL) in the absence or presence of IL-1β (1 ng/mL), IL-6 (100 ng/mL), and TNF-α (10 ng/mL).

### 4.5. MTT Cell Viability Assay

Primary culture chondrocytes were exposed to various concentrations of AJNR (10, 20, and 50 μg/mL) in the absence or presence of IL-1β (1 ng/mL), IL-6 (100 ng/mL), and TNF-α (10 ng/mL). 3-(4,5-Dimethylthiazol-2-yl)-2,5-diphenyltetrazolium bromide (MTT) assay was performed after 24 h of treatment. Briefly, the primary culture chondrocytes (5 × 10^4^) were seeded in 96-well plates. After 24 h of incubation, AJNR (20, 50, 100 μg/mL) was added to the cells in the absence or presence of IL-1β (1 ng/mL), IL-6 (100 ng/mL), and TNF-α (10 ng/mL). MTT solution was added to each well for 4 h at the final incubation time. After adding 100 μL of dimethyl sulfoxide solution, optical density was recorded using a microplate reader (570 nm).

### 4.6. Reverse Transcription-Polymerase Chain Reaction (RT-PCR)

Total RNA was extracted from primary cultured chondrocytes using TRIzol reagent (Molecular Research Center, Inc., Cincinnati, OH, USA). The quality and concentration of RNA were assessed using a NanoDrop™ 2000 spectrophotometer (Thermo Scientific, Waltham, MA, USA). The RNA was reverse transcribed, and the resulting cDNA was amplified by PCR or BIO-RAD Real-Time PCR system (CFX96™ Real-Time System, Bio-Health Materials Core-Facility, Jeju National University) using SYBR premixed Extaq reagent (Takara Bio, Mountain View, CA, USA). Target bands were quantified using the ImageJ densitometry software (NIH, Bethesda, MD). The PCR primers and experimental conditions are summarized in Appendix A. Glyceraldehyde-3-phosphate dehydrogenase (Gapdh) was used as an internal control.

### 4.7. Animal Model and AJNR Treatment

All experiments were approved by the Jeju National University Animal Care and Use Committee. The DMM-mouse model was established using 12-week-old C57BL/6J male mice (9 mice/group), as previously described [13,14,19]. The medial anterior meniscotibial ligament of the right knee was cut using a surgical scissor in the patellar tendon in the middle and tendon of the tibial plateau. In the sham operation group, only arthrotomy was performed without resection of the tibial ligament of the medial meniscus. The mice were administered intraperitoneal (IP) injection of AJNR (2 mg/kg) in 200 μL polyethylene glycol 400 (PEG-400) twice a week. The control group was injected with the same volume of PEG-400 reagent on the same schedule. The CIA mouse model was established using 7-week-old DBA/1J male mice (8 mice/group). RA was induced by type II collagen soluble in 0.05 M acetic acid at 4 °C. Emulsification was performed with an equal volume of Freund’s complete adjuvant and boosted with Freund’s incomplete adjuvant on day 21. Mice were administered AJNR (2 mg/kg) via IP injection with a newly formulated PEG-400. Control mice were injected with the same volume of PEG-400 reagent on the same schedule. The mice were examined every 3 days to measure the visual appearance of arthritis around the knee and ankle and to determine the corresponding severity and arthritis score [60,61].

### 4.8. Histological Analysis

The CIA (knee and ankle) or DMM (knee) samples were fixed with 4% paraffin formaldehyde for 24 h and then decalcified in 0.5 M EDTA solution at 4 °C for 4 weeks. Next, the samples were dehydrated using gradient ethanol and embedded in paraffin. Finally, the sample was sliced at a thickness of 5 mm and stained with Safranin-O/Fast Green for evaluation. Cartilage destruction and severity of synovitis were evaluated by experienced histological researchers who were blinded to the study groups. The Osteoarthritis Research Society International (OARSI) scoring system was used to evaluate cartilage degeneration, as previously described [38,62,63,64]. Subchondral plate thickness was measured using the Aperio Image Scope V12 software (Leica Biosystems, Buffalo Grove, IL, USA) [13,14,16].

### 4.9. Immunohistochemistry

The dehydrated slides were incubated with hydroperoxide (DACO LSAB 2 SYSTEM, HRP KIT; DAKO realTM), and then incubated for 45 min with trypsin at 37 °C. The slides were blocked with 1% bovine serum albumin for 60 min at room temperature. The sections were incubated with rabbit polyclonal antibody (4 μg/mL, ab52915; Abcam, Cambridge, UK) against Mmp3, rabbit polyclonal antibody (1:100 dilution, ab51072; Abcam) against Mmp13, and rabbit monoclonal antibody (1:200 dilution, #12912; Cell Signaling, Danvers, MA, USA) against IL-6 at 4 °C overnight. After incubation with anti-mouse or anti-rabbit secondary antibodies for 60 min, the slides were stained (Dako Real Envison™, Santa Clara, CA, USA).

### 4.10. Statistical Analysis

All statistical analyses were performed using IBM SPSS Statistics 24 (IBM Corp., Armonk, NY, USA). Experimental data were analyzed using the non-parametric Mann–Whitney U test and two-tailed Student’s *t*-tests with unequal sample sizes. Significance was accepted at a level of probability of 0.05 (*p* < 0.05). The results are shown as the mean ± standard error of the mean (SEM).

## 5. Conclusions

The present study showed that AJNR inhibited arthritis pathogenesis by reducing the expression of catabolic factors and enhancing the expression of anabolic factors without altering cell viability. Interestingly, AJNR exerted therapeutic effects in both OA and RA animal models. In particular, AJNR showed specific inhibitory effects against IL-6-mediated anabolic and catabolic imbalances. These findings indicate that AJNR has vast potential for the treatment of arthritis pathogenesis, which can lead to drug development.

## Figures and Tables

**Figure 1 pharmaceuticals-14-00776-f001:**
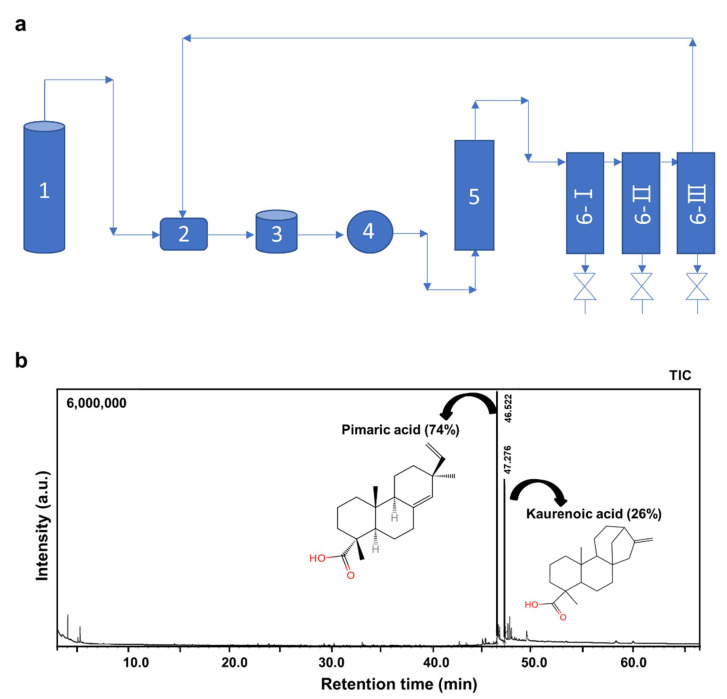
(**a**) Schematic drawing of the supercritical CO_2_ extraction device used in this study. CO_2_ source (1), a condenser (2), a CO_2_ storage tank (3), a CO_2_ pump (4), an extractor (5), and three separators (6). (**b**) Identification of the main compounds of AJNR extracted using the supercritical CO_2_ method.

**Figure 2 pharmaceuticals-14-00776-f002:**
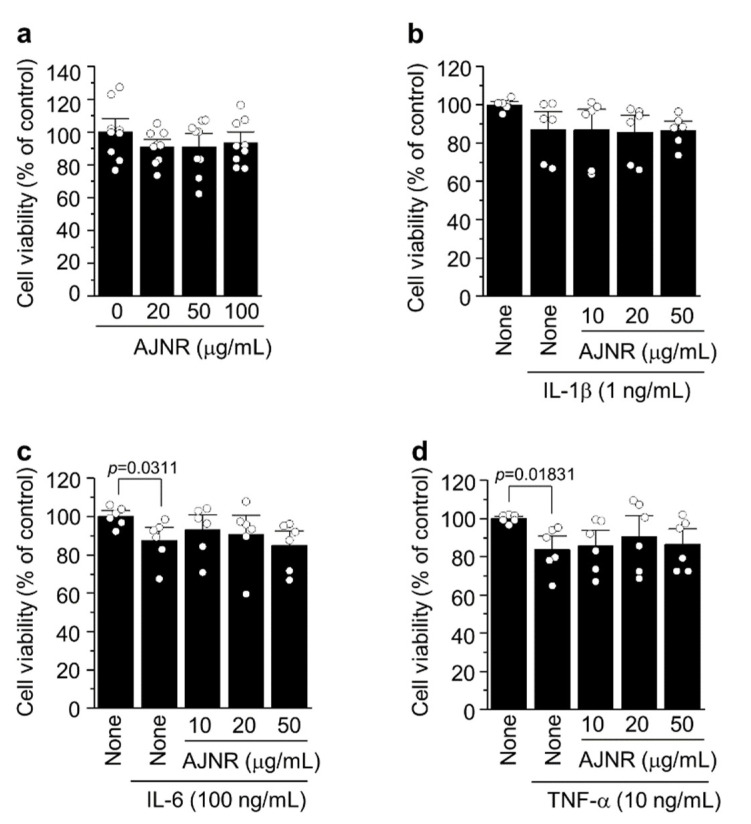
Effects of AJNR on the viability of primary cultured articular chondrocytes. (**a**) Primary cultured mouse articular chondrocytes were exposed to increasing concentration (0–100 µg/mL) of AJNR for 24 h and then subjected to MTT assay. The other groups were treated with AJNR (10, 20, 50 μg/mL) in the absence or presence of IL-1β (1 ng/mL) (**b**), IL-6 (100 ng/mL) (**c**), or TNF-α (10 ng/mL) (**d**), and then the cell viability of primary chondrocytes was evaluated by MTT assay. The results are representative of three independent experiments. Values are presented as the mean ± SEM and were evaluated using the two-tailed *t*-test.

**Figure 3 pharmaceuticals-14-00776-f003:**
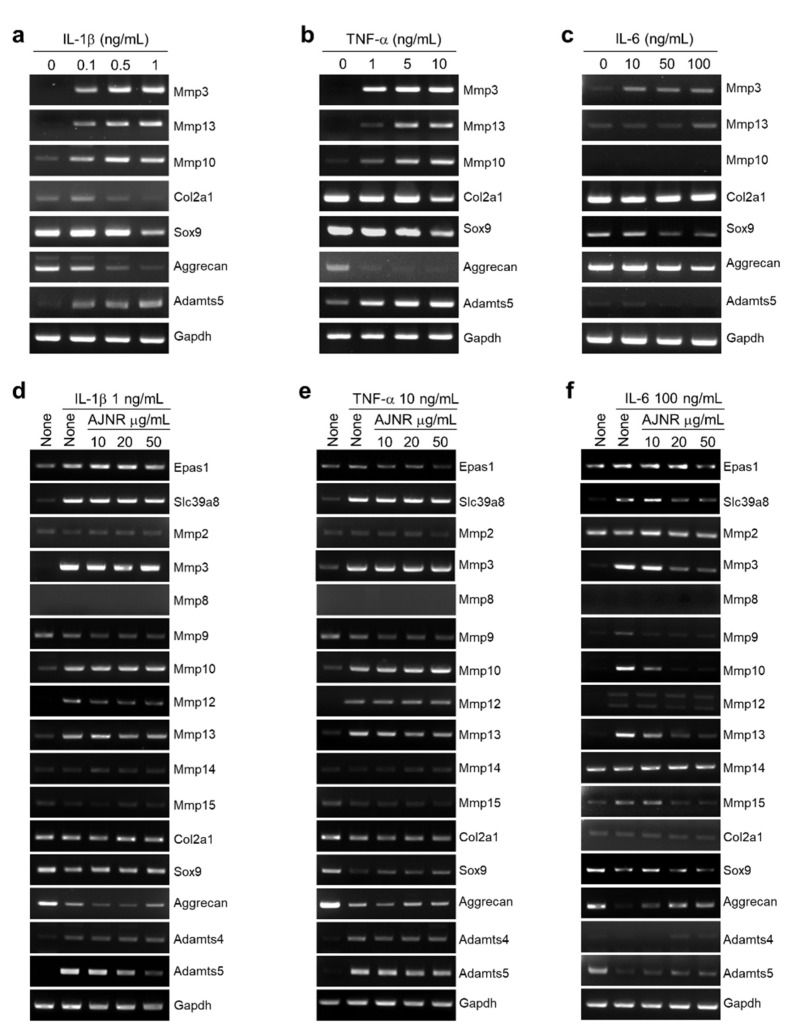
Inhibitory effects of AJNR on pro-inflammatory cytokine-induced catabolic expression in primary cultured chondrocytes. (**a**–**c**) Anabolic or catabolic factors in primary cultured mouse chondrocytes were analyzed following treatment with IL-1β, TNF-α, and IL-6. Chondrocytes were pre-incubated with the indicated concentrations of AJNR for 1 h and then stimulated with IL-1β (**d**), TNF-α (**e**), and IL-6 (**f**) for 24 h. RT-PCR analysis revealed the levels of anabolic and catabolic factors; semi-quantification data are presented in (**g**,**h**). The results are representative of three independent experiments from different pups. Values are presented as mean ± standard error of the mean (SEM) (* *p* < 0.05, ** *p* < 0.01, and *** *p* < 0.001), as analyzed via one-way ANOVA. Gapdh was used as an internal marker. AJNR, *Achyranthes japonica* Nakai Root; IL-1β, interleukin-1β; IL-6, interleukin 6; TNF-α, tumor necrosis factor-α; Epas1, endothelial PAS domain protein 1; Mmp2, -3, -8, -9, -10, -12, -13, -14, and -15, matrix metallo-proteinase-2, -3, -8, -9, -10, -12, -13, -14, and -15; Col2a1, collagen type II alpha 1 chain; Sox9, SRY-Box transcription factor 9; Adamts-4 and-5, ADAM metallopeptidase with thrombospondin type 1 motif-4 and -5; Gapdh, glyceraldehyde 3-phosphate dehydrogenase.

**Figure 4 pharmaceuticals-14-00776-f004:**
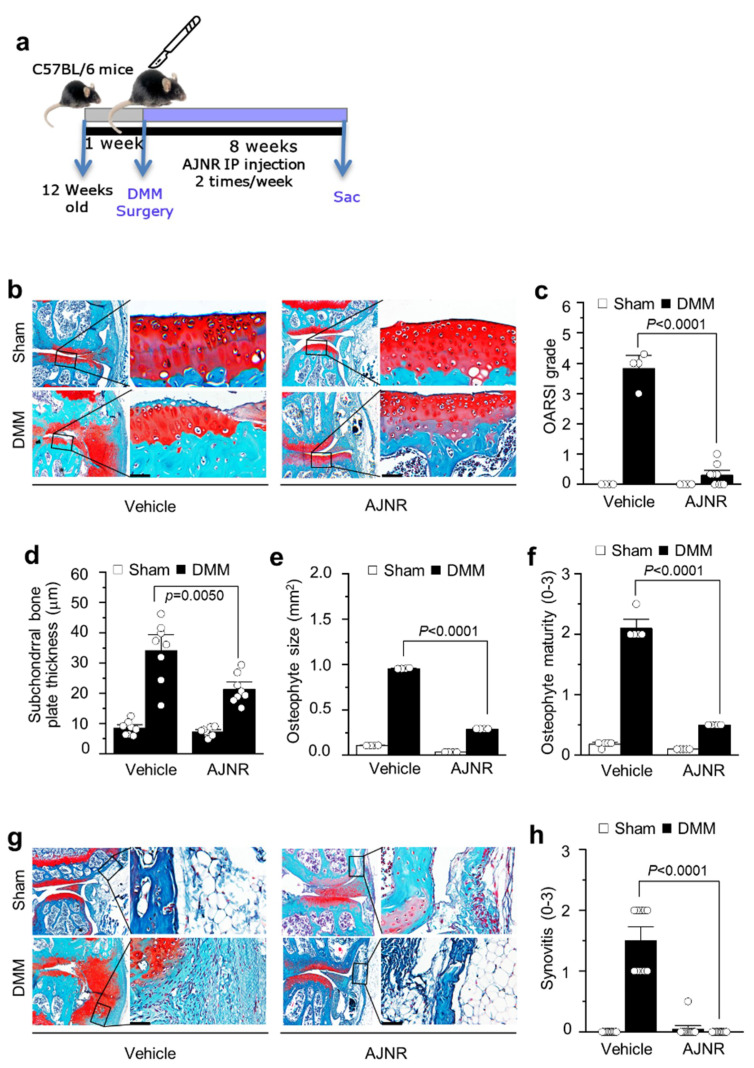
Inhibitory effects of AJNR in DMM-induced osteoarthritis mouse model. First, 12-week-old C57BL/6 mice were subjected to a sham operation or DMM surgery. Next, sham- or DMM-operated mice were administered AJNR (2 mg/kg) via intraperitoneal (IP) injection (2 times per week for 8 weeks). (**a**) The experiment scheme of AJNR in mouse DMM models. (**b**) Representative Safranin-O staining images showing the whole joint (40×), subchondral bone sclerosis, and cartilage destruction (400×). Osteoarthritis Research Society International (OARSI) grade (for cartilage destruction) (**c**), subchondral bone plate thickness (for subchondral bone sclerosis) (**d**), osteophyte size (**e**), and osteophyte maturity (**f**) were quantified (*n* ≥ 4 mice per group). (**g**) Representative Safranin-O and H&E staining images of synovial inflammation, in sham, DMM, and AJNR-treated DMM mice. Scores of synovial inflammation (**h**) is presented (*n* = 12 mice per group). The results are representative of three independent experiments. Values are presented as the mean ± SEM and were evaluated using the Mann–Whitney U test. Scale bar: 50 μm.

**Figure 5 pharmaceuticals-14-00776-f005:**
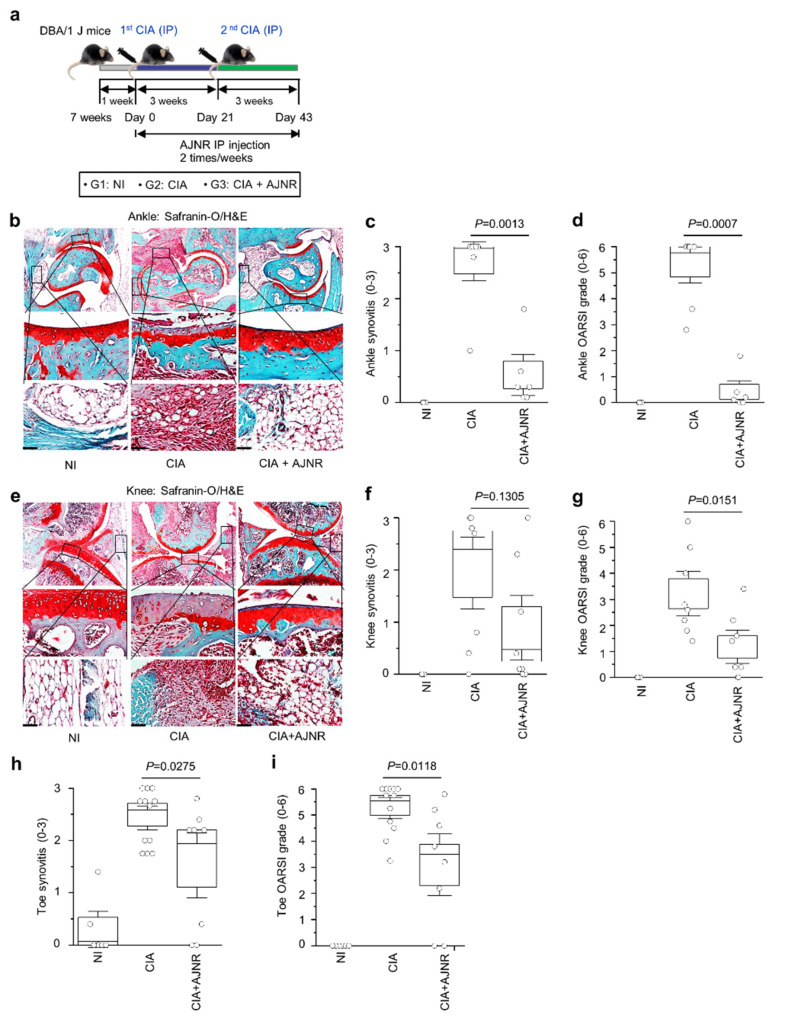
Inhibitory effects of AJNR against collagenase-induced arthritis (CIA)-induced cartilage degeneration and synovitis. First, CIA models were established using 10-week-old DBA/1 J mice. During arthritis induction, AJNR intraperitoneal (IP) injections were administered (2 times per week for 6 weeks). (**a**) The scheme of AJNR treatment in the CIA mouse model. (**b**) Representative Safranin-O and hematoxylin and eosin stain (H&E) staining images of ankle cartilage degeneration and synovial inflammation, respectively, in the non-immunized (NI), CIA, and AJNR-treated CIA mice. Scores of synovial inflammation (**c**) and Osteoarthritis Research Society International (OARSI) (**d**) in the ankle are presented (*n* ≥ 4 mice per group). (**e**) Representative Safranin-O and H&E staining images of knee cartilage degeneration and synovial inflammation, respectively, in NI, CIA, and AJNR-treated CIA mice. Scores of synovial inflammation (**f**) and OARSI (**g**) in the knee are presented (*n* ≥ 4 mice per group). Scores of toe synovitis (**h**) and OARSI (**i**) are also presented (*n* ≥ 6 mice per group). The results are representative of three independent experiments. Values are presented as the mean ± SEM and were evaluated using the Mann–Whitney U test. Scale bar: 50 μm.

**Figure 6 pharmaceuticals-14-00776-f006:**
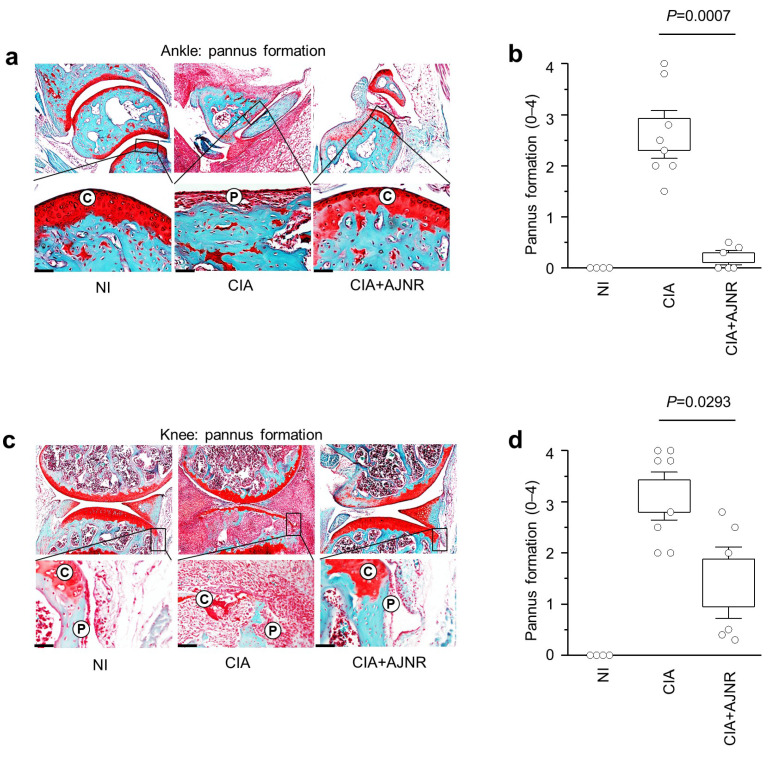
Inhibitory effects of AJNR in CIA-induced pannus formation. Collagenase-induced arthritis (CIA) models were established using 7-week-old DBA/1 J mice. During arthritis induction, AJNR intraperitoneal injections were administered (2 times per week for 6 weeks). (**a**) Representative Safranin-O and H&E staining images of the ankle pannus of non-immunized (NI), CIA, and AJNR-treated CIA mice. Score of ankle pannus formation is presented in (**b**) (*n* ≥ 4 mice per group). (**c**) Representative Safranin-O and H&E staining images of the knee pannus of NI, CIA, and AJNR-treated CIA mice. Score of knee pannus formation is presented in (**d**) (*n* ≥ 4 mice per group). The results are representative of three independent experiments. Values are presented as the mean ± SEM and were evaluated using the Mann–Whitney U test. Scale bar: 50 μm.

**Figure 7 pharmaceuticals-14-00776-f007:**
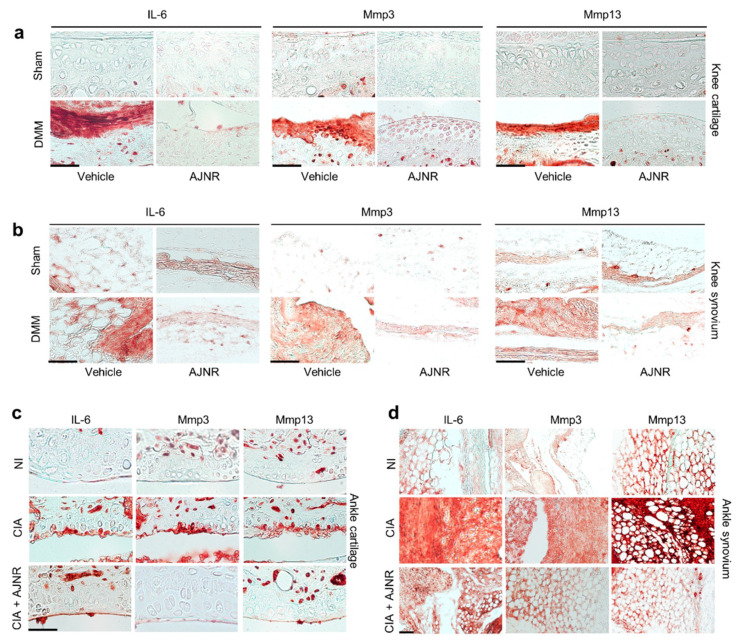
(**a**,**b**) AJNR inhibits IL-6-mediated Mmp3 and Mmp13 expression in destabilization of the medial meniscus (DMM) surgery-induced osteoarthritis. First, 12-week-old C57BL/6 male mice were subjected to a sham operation or DMM surgery. Sham- or DMM-operated mice were intraperitoneally injected with AJNR (2 mg/kg) (2 times per week for 8 weeks). (**a**) Representative images of IL-6, Mmp3, and Mmp13 immunostaining in the knee cartilage tissue sections of sham, DMM, and AJNR-treated DMM mice (*n* = 4 mice per group). (**b**) Representative images of IL-6, Mmp3, and Mmp13 immunostaining in the synovial tissue sections of sham, DMM, and AJNR-treated DMM mice (*n* = 4 mice per group). (**c**,**d**) AJNR inhibits IL-6-mediated Mmp3 and Mmp13 expression in CIA-induced rheumatoid arthritis. CIA models were established using 7-week-old DBA/1J male mice. During arthritis induction, AJNR intraperitoneal injections were administered (2 times per week for 6 weeks). (**c**) Representative IL-6, Mmp3, and Mmp13 immunostaining in the ankle cartilage tissue sections of non-immunized (NI), CIA, and AJNR-treated CIA mice (*n* = 4 mice per group). (**d**) Representative images of IL-6, Mmp3, and Mmp13 immunostaining in the synovial tissue sections of NI, CIA, and AJNR-treated CIA mice (*n* = 4 mice per group). The results are representative of three independent experiments. Scale bar: 50 μm.

## Data Availability

Data is contained within the article and Appendix A.

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
