# Peer review of "Inhibitory Effects of IL-6-Mediated Matrix Metalloproteinase-3 and -13 by Achyranthes japonica Nakai Root in Osteoarthritis and Rheumatoid Arthritis Mice Models"

_pharmaceuticals, 2021, doi:10.3390/ph14080776_

Round 1
Reviewer 1 Report
Authors proposed a paper entitled “Supercritical CO2 extract of Achyranthes japonica Nakai root inhibits arthritis pathogenesis in vitro and in vivo” for the publication in pharmaceuticals, mdpi.
Although the paper has a quite good scientific soundness, it requires to be reviewed before it is accepted for publication.
Here is the list of my comments and opinions:
I suggest adding an abbreviation list to this paper, according to this Journal guidelines.
Line 24. In the abstract, at this line, the authors say that the extraction has been performed using carbon dioxide in supercritical conditions. however, is it possible to also define the kind of supercritical assisted process employed in this manner?
Line 30. “in in vitro”. I would say “related to in vitro”
Introduction. In my opinion, the introduction section lacks of any information. The title talks about supercritical assisted process, but no description of these processes neither a comparison with conventional processes has been proposed or performed.
Line 224. Authors cannot just talk of effect of supercritical CO2. CO2 is at supercritical state, but a comparison among operating temperatures, pressures, gas to liquid ratios is not present.
Mean dimensions of natural matter has not been investigated or commented.
Line 272. “by extracting raw AJNR using a supercritical CO2,”. What are the authors talking about? Which process? “using a supercritical CO2” does not identify the process. How did they reach these conditions?
Where are the expression of the pressure and temperature operating conditions of the process? these conditions are given in the subsequent lines, but I think that should be moved to the top of this section. Also, I suggest adding a sketch of the process, describing it in details and providing proper further references. This should not be considered as supplementary material. Nowadays, the processes for the production of pharmaceutical formulations are particularly important. It is worth citing the differences among conventional and supercritical assisted processes, that are significantly making the difference.
“CO2 source, a condenser, a CO2 276 storage tank, a CO2 pump, an extractor, and three separators”. there is no information about model, manufacturer and country of these elements of the plant. authors just cannot say that they bought this plant and used it as it is, in my opinion.
Line 289. Did you check the solvent residue in the final extract, before and after rotary evaporation refining step?
Authors do not explain how the CO2 is released to the atmosphere after the extraction process.
Conclusions should be improved with more critical comments on the results obtained.
Author Response
Comments and Suggestions for Authors
Reviewer #1:
Authors proposed a paper entitled “Supercritical CO2 extract of Achyranthes japonica Nakai root inhibits arthritis pathogenesis in vitro and in vivo” for the publication in pharmaceuticals, mdpi.
Although the paper has a quite good scientific soundness, it requires to be reviewed before it is accepted for publication.
Here is the list of my comments and opinions:
Comment 1.
I suggest adding an abbreviation list to this paper, according to this Journal guidelines.
[Response]
The journal requires no abbreviation list. As per the journal guidelines, we have defined all abbreviations the first time they appear in the Abstract and Main Text.
Comment 2.
Line 24. In the abstract, at this line, the authors say that the extraction has been performed using carbon dioxide in supercritical conditions. however, is it possible to also define the kind of supercritical assisted process employed in this manner?
[Response]
In the Materials and Methods section, we have described in detail the carbon dioxide supercritical fluid conditions, e.g., the extraction pressure (40 MPa) and temperature (60 °C), each separator’s pressure and temperature, and CO2 flow rate (page 12, line 355~358).
Comment 3.
Line 30. “in in vitro”. I would say “related to in vitro”
[Response]
In accordance with the reviewer’s comment, we have edited this sentence (page 1, line 31~32).
Comment 4.
Introduction. In my opinion, the introduction section lacks of any information. The title talks about supercritical assisted process, but no description of these processes neither a comparison with conventional processes has been proposed or performed.
[Response]
- In accordance with the reviewer’s comment, we have added the information of carbon dioxide supercritical fluid in the Introduction section (page 2, line 81~90).
[In recent years, supercritical fluid extraction has received a great deal of attention in the pharmaceutical industry [28,29]. The supercritical fluid technology has been applied to the production of food, flavoring, nutrients, and bioactive components from plant sources [30,31]. CO2 is the most ideal solvent for the supercritical extraction of natural products, as supercritical CO2 has low critical parameters as well as hydrophobic and nonpolar characteristics [32]. Moreover, the ambient critical temperature makes it suitable for the extraction of thermolabile component without degradation [32]. Furthermore, the CO2 supercritical fluid extraction method has been recognized as an environmentally friendly technology because it is completely free from the use of potentially toxic chemical solvents [28,29].]
- It is difficult to compare supercritical fluid technology (SCF) with conventional processes because we aimed only to obtain extracts using this technology. Our purpose was not to compare SCF with other methods and not to compare samples obtained at different conditions either. The novelty of this study is that the extracts obtained by SCF showed some activities that have not been reported by previous studies. However, we did not examine the effect of the extraction conditions on the compositions, which we plan to investigate in the next phase of research
Comment 5.
Line 224. Authors cannot just talk of effect of supercritical CO2. CO2 is at supercritical state, but a comparison among operating temperatures, pressures, gas to liquid ratios is not present.
[Response]
In the Materials and Methods section, we have described in detail the carbon dioxide supercritical fluid parameters, e.g., the extraction pressure (40 MPa) and temperature (60 °C); the pressure and temperature of each separator: 8 MPa, 50 °C (separators 6-I) and 5 MPa, 40 °C (separators 6-II and 6-III), respectively; CO2 flow rate controlled at 20 kg/h; and extraction time of 2 h (page 12, line 355~358).
Comment 6.
Mean dimensions of natural matter has not been investigated or commented.
[Response]
We have mentioned the size of the raw materials (approximately 40 meshes) in the Material and Methods section (page 12, line 351).
Comment 7.
Line 272. “by extracting raw AJNR using a supercritical CO2,”. What are the authors talking about? Which process? “using a supercritical CO2” does not identify the process. How did they reach these conditions?
[Response]
We have edited this sentence as follows: “The extracts for the experiment were prepared using supercritical CO2 extraction” (page 12, line 344)
Comment 8.
Where are the expression of the pressure and temperature operating conditions of the process? these conditions are given in the subsequent lines, but I think that should be moved to the top of this section. Also, I suggest adding a sketch of the process, describing it in details and providing proper further references. This should not be considered as supplementary material. Nowadays, the processes for the production of pharmaceutical formulations are particularly important. It is worth citing the differences among conventional and supercritical assisted processes, that are significantly making the difference.
[Response]
As we mentioned before, we have described the carbon dioxide supercritical fluid conditions in detail in the Materials and Methods section. In accordance with the reviewer’s recommendation, we have moved the schematic drawing of the supercritical CO2 extraction device from supplementary materials to Fig. 1 of the main text (page 3).
Comment 9.
“CO2 source, a condenser, a CO2 276 storage tank, a CO2 pump, an extractor, and three separators”. there is no information about model, manufacturer and country of these elements of the plant. authors just cannot say that they bought this plant and used it as it is, in my opinion.
[Response]
We have mentioned the model of the devices in the Material and Methods section (page 12, line 347)
Comment 10.
Line 289. Did you check the solvent residue in the final extract, before and after rotary evaporation refining step?
[Response]
We completely removed the solvent in a rotary evaporation refining step. Moreover, even though the extract contained solvent residue, this might not be a problem because we dissolved the extract with the same solvent used in SCF.
Comment 11.
Authors do not explain how the CO2 is released to the atmosphere after the extraction process. [Response]
We have added this sentence in the Material and Methods (page 12 line 360): “CO2 was released to the atmosphere via an empty valve.”
Comment 12.
Conclusions should be improved with more critical comments on the results obtained.
[Response]
We have revised the Conclusions section according to the reviewer’s suggestion (page 11, line 275~279, line 295~301, line312~314; page 12, line 321~334)
Reviewer 2 Report
The authors investigated the effect of Achyranthes japonica Nakai root (AJNR) extract prepared by supercritical CO2 on mouse chondrocytes and RA and OA rodent model. Some data should be semiquantified, particularly, to support the conclusion. Immunohistology for the OA model is not shown. The discussion is very superficial. some information in the method section are lacking.
It could not find the information in the abstract that mice models were used and also not that the chondrocyte cultures used derive from mice. The title should be improved, please include OA and RA, MMPs and IL-6 (see conclusion of the abstract) to show that it is different from other studies (the authors cited other works about AJNR and arthritis (line 72-73))
line 70: (style) 2x "compound" in one sentence, please substitute one time by "component"
line 77: extracted using supercritical CO2, what is the efficacy of this strategy in contrast to other extraction procedures?
line 80 "con-firm" should be "confirm"
line 90: "cell viability" write "chondrocyte viability"
figure 2: line 99: add "articular"
2.3 "key OA inducer genes" these genes should be mentioned in the introduction since they are so far not well known and need explanation
line 111: "Mmp": write it generally in capital letters
Figure 3: how many independent experiments were conducted? please try to semiquantify the results from all experiments (densitometry)
The cytokines used (TNF, IL-6, IL-1) did they derive from the species mouse?
Figure 4: subchondral plate thickness: how was it measured?
IP injection, why could it not be administered per os?
Please discuss the selected concentrations
Figure 7: why was the immunhistology only shown for the RA model and not for OA
Discussion
lines 224-226. provide a reference for the first sentence. post-traumatic OA model is not included in the present study.
lines 251-251: are all these components not extracted by supercritical CO2?
4.5 provide the source and species of cytokines used.
line 329: male or female mice? (OA/CIA model)
"nine mice" / "eight mice" per group or at all?
provide number of independent experiments for each method.
line 330: the sentence makes no sense - rewrite. write "medial anterior meniscotibial ligament"
Author Response
Reviewer #2:
Comment 1.
The authors investigated the effect of Achyranthes japonica Nakai root (AJNR) extract prepared by supercritical CO2 on mouse chondrocytes and RA and OA rodent model. Some data should be semiquantified, particularly, to support the conclusion. Immunohistology for the OA model is not shown. The discussion is very superficial. some information in the method section are lacking.
[Response]
- We have provided semiquantitative data in Fig. 3 (page 5)
- Immunohistochemistry analysis results for OA have been provided in Fig. 7 (page 10)
- We have revised the Discussion section (page 11, line 275~279, line 295~301, line312~314; page 12, line 321~334)
- We have provided more information in the Methods section, e.g., cytokine information, supercritical CO2 device information, mouse sex and number, and some software programs used in this study (page 12, line 340-342, 347, 360; page 13, line 397-398, 403-411; page 14, line 427-429)
Comment 2.
It could not find the information in the abstract that mice models were used and also not that the chondrocyte cultures used derive from mice. The title should be improved, please include OA and RA, MMPs and IL-6 (see conclusion of the abstract) to show that it is different from other studies (the authors cited other works about AJNR and arthritis (line 72-73)).
[Response]
- The mice models used in this study are described in the Abstract (destabilization of the medial meniscus (DMM) as OA model and collagenase-induced arthritis (CIA) as RA model) (page 1, line 28~29).
- The chondrocyte culture method is also well described in the Material and Methods section (4.4 Primary culture of mouse knee joint chondrocytes and treatment with AJNR) (page 13, line 370~379).
- In accordance with the reviewer suggestion, we have revised the title from “Supercritical CO2 extract of Achyranthes japonica Nakai root inhibits arthritis pathogenesis in vitro and in vivo” to “Inhibitory effects of IL-6-mediated matrix metalloproteinase-3 and -13 by Achyranthes japonica Nakai root in osteoarthritis and rheumatoid arthritis mice models.” (page 1, line 2~4)
Comment 3.
line 70: (style) 2x "compound" in one sentence, please substitute one time by "component" [Response]
We have edited the sentence (page 2, line 91~92).
Comment 4.
line 77: extracted using supercritical CO2, what is the efficacy of this strategy in contrast to other extraction procedures?
[Response]
It is difficult to illustrate the efficacy of supercritical CO2 extraction compared with other extraction methods because the extract components and ratios differ depending on the pressure, temperature, and co-solvent used. Nevertheless, extraction method comparison was not the aim of this study. The key point of the study is that using supercritical CO2 extraction under our extraction conditions, we discovered two main compounds (pimaric acid and kaurenoic acid) of AJNR, which we believe are the active compounds of AJNR.
Comment 5.
line 80 "con-firm" should be "confirm"
[Response]
We have edited the word (page 3, line 101).
Comment 6.
line 90: "cell viability" write "chondrocyte viability"
[Response]
We have edited the words accordingly (page 3, line 119).
Comment 7.
figure 2: line 99: add "articular"
[Response]
We have added the word (page 4, line 137).
Comment 8.
2.3 "key OA inducer genes" these genes should be mentioned in the introduction since they are so far not well known and need explanation
[Response]
We have described the key OA inducer genes, namely Epas1, Slc39a8, Esrrg, Nampt, Mtf1, RUNX2, BATF, and RORα, in the Introduction section (page 2, line 59~68).
[It has been reported that some particular transcription factors upregulate pro-inflammatory cytokines, e.g., endothelial PAS domain protein 1 (Epas1) [2], estrogen-related receptor gamma (Esrrg) [14], nicotinamide phosphoribosyl-transferase (Nampt) [15], metal regulatory transcription factor 1 (Mtf1) [16], runt-related transcription factor 2 (RUNX2) [17], basic leucine zipper transcription factor, ATF-like (BATF) [18], and RAR-related orphan receptor α (RORα) [19]. Previous studies indicated that HIF-2 is a central player in OA pathogenesis, and that its target transcription factors or non-transcription factors are positive regulators of catabolic factors in OA pathogenesis [15,20,21]. In addition, toll-like receptor (TLR) signaling [22] as well as arginine [23], selenium [24], and cholesterol metabolisms [19] are associated with pro-inflammatory cytokines in OA pathogenesis.]
Comment 9.
line 111: "Mmp": write it generally in capital letters
[Response]
We have edited it accordingly (page 4, line 131).
Comment 10.
Figure 3: how many independent experiments were conducted? please try to semiquantify the results from all experiments (densitometry)
[Response]
- We performed three independent experiments using different pups (we have described this in the figure legends) (page 6, line 156-157).
- We have added semiquantitative results of key catabolic factors (Mmp3, Mmp13, and Adamts 5) and anabolic factors (Col2a1, Sox9, and Aggrecan) (page 5).
Comment 11.
The cytokines used (TNF, IL-6, IL-1) did they derive from the species mouse?
[Response]
We used recombinant human IL-6 (PHC0064) from Gibco as well as recombinant mouse TNF-a (Cat.No. Z02918-20) and human IL-1b (Cat.No. Z02922-10) from GeneScript. These cytokines are already confirmed and used widely in chondrocytes of mouse origin (Nature. 2019 Feb;566(7743):254-258, Cell. 2014 Feb 13;156(4):730-43, Nat Commun. 2017 Dec 15;8(1):2133, Nat Commun. 2019 Jan 8;10(1):77, Nat Med. 2010 Jun;16(6):687-93). We have added information of these cytokines in the Material and Methods section (page 12, line 340~342).
Comment 12.
Figure 4: subchondral plate thickness: how was it measured?
[Response]
Subchondral plate thickness was measured using Aperio Image Scope V12 (Leica Biosystems). We measured 300 µm width calcified cartilage and measured bone marrow distance from the calcified cartilage. Based on the calculated area, subchondral plate thickness was obtained. We have described this in detail in the Material and Method section (page 14, line 427~429).
Comment 13.
IP injection, why could it not be administered per os?
[Response]
We performed IP injection to observe the effects of AJNR on arthritis mice model, with an observation period of 8 weeks for the osteoarthritis model and 7 weeks for the rheumatoid arthritis model. We chose the IP administration route because the effects of drugs administered via IP injection should usually appear within a short time compared with those administered orally. Moreover, we think that oral AJNR will affect our osteoarthritis and rheumatoid arthritis mouse models.
Comment 14.
Please discuss the selected concentrations
[Response]
We have discussed the selected concentrations for the in vitro and in vivo experiments in the Discussion section (page 11, line 275~279).
Comment 15.
Figure 7: why was the immunhistology only shown for the RA model and not for OA
[Response]
We presented immunohistochemistry data for OA in the Supplementary information. However, in consideration of the reviewer’s comment, we have moved the immunohistochemistry data of OA in the Supplementary information to Fig 7a and b in the Main Text (page 10).
Discussion
Comment 16.
lines 224-226. provide a reference for the first sentence. post-traumatic OA model is not included in the present study.
[Response]
DMM model is a well-known post-traumatic OA model in the OA field.
Comment 17.
lines 251-251: are all these components not extracted by supercritical CO2?
[Response]
The supercritical CO2 extraction method can extract flavonoids, alkaloids, allantoin, succinic acid, and β-sitosterol with the addition of a co-solvent. As discussed previously, the extracted compound is dependent on the pressure, temperature, and co-solvent used.
Comment 18.
4.5 provide the source and species of cytokines used.
[Response]
We have added information of the cytokines in the Materials and Methods section (4.1) (page 12, line 340~342)
Comment 19.
line 329: male or female mice? (OA/CIA model).
[Response]
We used male mice in all experiments. We have added the mouse sex information in the Materials and Methods section (page 13, line 403 and line 411).
Comment 20.
"nine mice" / "eight mice" per group or at all? .
[Response]
We used 9 mice/group for the DMM model and 8 mice/group for the CIA model. We have added this information in the Materials and Methods section (page 13, line 404 and line 411).
Comment 21.
provide number of independent experiments for each method.
[Response]
We have provided the number of independent experiments in all figure legends (page 4, line 141; page 6, line 157; page 7, line 183, page 8, line 200; page 9, line 222; page 11, line 271).
Comment 22.
line 330: the sentence makes no sense - rewrite. write "medial anterior meniscotibial ligament".
[Response]
We have edited the sentence accordingly (page 13, line 404).
Round 2
Reviewer 1 Report
Authors provided an amended version of the paper.
The paper has been improved, after responding point by point to the issues raised by the reviewers.
Additional information has been added to the introduction.
A quite long bibliographic study has been performed in this paper.
Results have a good scientific soundness.
I only suggest adding an abbreviation list, according to the guidelines of this Journal.
Maybe the paragraph about conclusions could go into a separate section.
Then, the paper deserves to be published.
Author Response
Comments and Suggestions for Authors
Reviewer:
Authors provided an amended version of the paper.
The paper has been improved, after responding point by point to the issues raised by the reviewers.
Additional information has been added to the introduction.
A quite long bibliographic study has been performed in this paper.
Results have a good scientific soundness.
Comment 1.
I only suggest adding an abbreviation list, according to the guidelines of this Journal.
[Response]
According to the reviewer’s comment, we have added an abbreviation list after the “Conflict of interest” (page 14, lines 460~487).
Comment 2.
Maybe the paragraph about conclusions could go into a separate section.
[Response]
According to the reviewer’s comment, we have relocated the conclusions at the end of the “Material and Methods” section (page 14, lines 440~447).
- Conclusions
Then, the paper deserves to be published.
Editor:
Comment 1.
(1) Reviewer 1 is right. According to our format, please add Conclusions
part, you can move the last paragraph "In conclusion,...." of Discussion to
"5. Conclusions" part.
[Response]
According to the reviewer and editor, we have relocated the conclusions at the end of the “Material and Methods” section (page 14, lines 440~447).
- Conclusions
Comment 2.
(2) We also encourage authors to add an abbreviation list after the "Conflict
of interest" like reviewer 1 suggestion.
[Response]
According to the reviewer and editor comment, we have added an abbreviation list after the “Conflict of interest” (page 14, lines 460~487).
